

# Potential role of ghrelin in neuroprotection and cognitive function: implications for diabetic cognitive impairment

Yuhan Zhang[1], Ruihua Zhang[1], Xin Wang[1], Leilei Shi[1], Hongzhe Zhu[1] and Jiping Liu[1,2,3,4]

[1] Department of Pharmacology, Shaanxi University of Chinese Medicine, Xi'an, Xianyang, China
[2] Key Laboratory of Pharmacodynamic Mechanism and Material Basis of Traditional Chinese Medicine, Shaanxi Administration of Traditional Chinese Medicine, Xianyang, China
[3] Shaanxi Key Laboratory for Safety Monitoring of Food and Drug, Xianyang, China
[4] Engineering Research Center of Brain Health Industry of Chinese Medicine, Universities of Shaanxi Province, Xianyang, China

## ABSTRACT

Ghrelin is a class of brain and intestinal peptides. It regulates food intake and body glucose levels and maintains cellular homeostasis. In recent years, research has revealed that ghrelin may positively impact learning and memory. Despite ghrelin's multiple functions in the central nervous system, its use as a therapeutic agent for neurologic dysfunction remains unclear. Diabetic cognitive impairment (DCI) is a severe neurological complication of diabetes mellitus. Its incidence is increasing as a comorbidity in endocrinology and neurology. Additionally, it is a risk factor for Alzheimer's disease (AD). Ghrelin levels are altered in patients with diabetes mellitus combined with cognitive impairment. Furthermore, modulation of ghrelin levels improved cognitive function in rats with DCI. These findings suggest the potential therapeutic importance of ghrelin in the pathogenesis of DCI. This article presents a comprehensive review of the pathogenesis of DCI and its potential modulation by ghrelin and its mimics. Furthermore, this study elucidates the therapeutic prospects of ghrelin and its mimics for DCI, aiming to identify novel therapeutic targets and research avenues for the prevention and management of DCI in the future.

## INTRODUCTION

With the aging population and changes in dietary structure and lifestyle, the prevalence of diabetes mellitus (DM) is increasing annually. It is expected to continue rising to 10.2% by 2030. Therefore, how to prevent and control diabetes mellitus and its complications has become a major issue that needs to be addressed in the future. Diabetic cognitive impairment (DCI) represents a neurological complication observed in individuals with diabetes. This condition is primarily characterized by cognitive dysfunction, diminished learning capacity, and impaired decision-making abilities. Epidemiological surveys have

Corresponding author
Jiping Liu, ljp0711@sntcm.edu.cn

indicated that DCI is a noncommunicable disease affecting nearly 400 million individuals (*Ehtewish, Arredouani & El-Agnaf, 2022*). As the disease progresses, the patient's self-care ability will be impaired. In severe cases, it may even evolve into Alzheimer's disease (AD) (*Zhang et al., 2023b*), imposing a burden on families, society, and even the country. Hence, most scholars have thoroughly examined and deliberated upon the correlation between DCI and AD. The prevention of further cognitive impairment must be considered a pivotal aspect of current DCI treatment. The pathophysiological mechanism of DCI remains elusive, and the therapeutic drugs currently employed in clinical practice merely serve to delay disease progression. Consequently, scholars must explore and experiment with safe and efficacious therapies for DCI.

Ghrelin is a brain-gut peptide that regulates insulin sensitivity and is produced by X/A-like cells in rats and P/D1 cells in humans (*Barazzoni et al., 2008*; *Poher, Tschöp & Müller, 2018*). The structure of ghrelin comprises 28 amino acids with a structural formula of $C_{149}H_{249}N_{47}O_{42}$. Its molecular weight is approximately 3,300 Da. *In vivo*, it predominantly exists in two forms: acylated and nonacylated ghrelin. Both forms can enter the central nervous system. Acylated ghrelin refers to the third amino acid (serine) site of the polypeptide chain that is acylated by the action of ghrelin-acyltransferase (GOAT) to become the predominantly active form *in vivo*. Only acylated ghrelin serves as an endogenous ligand for GHS-R1a, while the receptor for deacylated growth hormone-releasing peptides remains unknown. Ghrelin can reach the central nervous system through two pathways: one involves direct binding to the GHS-R1a receptor in the stomach, transmitting signals to the hypothalamus *via* the vagus nerve; the other pathway involves crossing the blood–brain barrier and entering various regions of the brain through circulation, where it exerts its effects (*Perelló et al., 2022*; *Rhea et al., 2018*). In its acylated form, ghrelin regulates food intake, modulates energy homeostasis, and provides cardioprotection (*Mani, Shankar & Zigman, 2019*) (Fig. 1). In addition, ghrelin plays a significant role in the pathogenesis of type 2 diabetes. Moreover, ghrelin binds to the GHS-R and exhibits neuroprotective effects, including reducing Aβ deposition, improving impaired neurofactor metabolism, and mitigating mitochondrial dysfunction (*Mao et al., 2024*; *Tian, Wang & Du, 2023*). The suggestion is that ghrelin may serve as a potential link between diabetes mellitus and nerve damage, exhibiting a significant correspondence to the pathogenesis of DCI. Nonetheless, the investigation into the association between ghrelin and DCI remains confined mainly to foundational experimental studies. Consequently, this article aims to offer a comprehensive review and forward-looking analysis of the potential role of ghrelin in the prevention and treatment of DCI, as well as the developmental prospects of ghrelin mimetics. This work will likely contribute to identifying novel targets and innovative strategies for managing DCI.

## THE RELEVANCE OF GHRELIN AND ITS RECEPTOR IN DCI

### Ghrelin

Subsequent clinical studies and *in vitro* and *in vivo* experiments further substantiated the correlation between ghrelin and DCI. The levels of ghrelin were significantly lower in patients with diabetes combined with cognitive impairment than in normal subjects (*Sang*

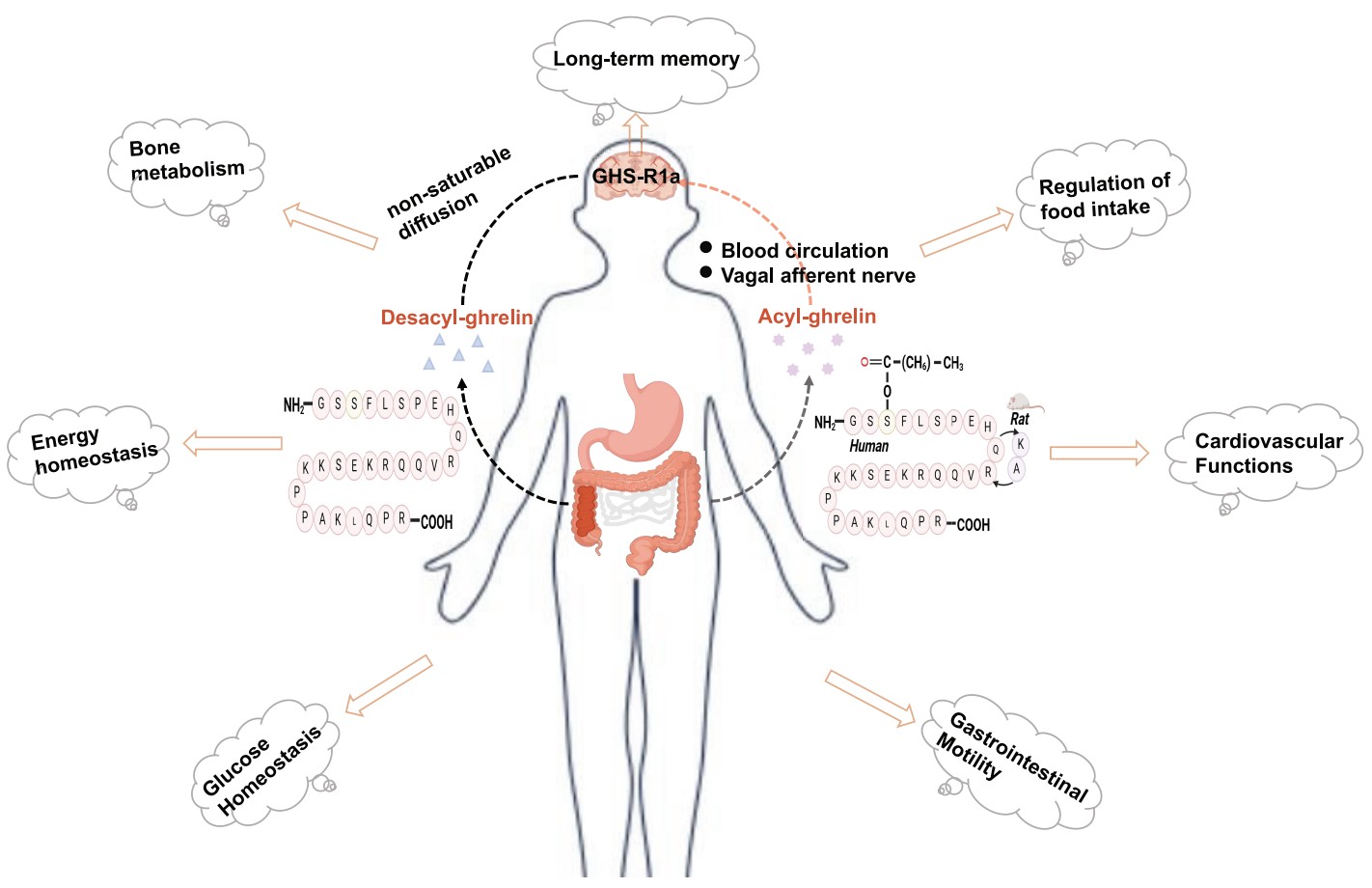

**Figure 1 Physiologic effects of ghrelin.** Created in Biorender.

*et al., 2018*). Further logistic regression analyses demonstrated that ghrelin levels were an independent factor for mild cognitive impairment (MCI) in patients with type 2 diabetes mellitus (T2DM), additionally, partial correlation analyses indicated that growth hormone-releasing peptide levels were positively correlated with delayed recall scores on the Montreal Cognitive Assessment and the Auditory Verbal Learning Test (*Huang et al., 2017*). The Wisconsin Card Sorting Test identified ghrelin as a potential new predictor associated with executive ability in T2DM patients (*Chen et al., 2017*). This finding has been corroborated through both animal and cellular experiments (Table 1). In conclusion, ghrelin may be involved in the onset and development of DCI; however, the impact of ghrelin on pathological changes in DCI remains insufficiently investigated and warrants further exploration.

## Growth hormone secretagogue receptor

The growth hormone secretagogue receptor (GHS-R) is a G protein-coupled receptor consisting of seven transmembrane helical structural domains, encompassing 366 amino acids (*Engelstoft et al., 2013*). It exhibits abundant expression in various brain regions, particularly in the hippocampus. GHS-R has two splice variants, GHS-R1a and GHS-R1b. GHS-R1a interacts with ghrelin and activates various downstream signalling pathways,

**Table 1 The association of ghrelin with diabetes cognitive impairment-related pathologies.**

| Mechanism of action | Experimental model | Main result |
|---|---|---|
| Apoptosis in hippocampal neurons | Following the establishment of the DM rat, *in vivo* Ghrelin and [D-Lys(3)]-GHRP-6 were administered. | Ghrelin decreases the number of apoptotic neurons in the hippocampus, exhibits significant anti-inflammatory effects, and enhances cognitive function. These effects were inhibited by [D-Lys (3)]-GHRP-6 (*Ma et al., 2011*). |
| Neurodegeneration | | Ghrelin modulates MMP levels, inhibits neuroinflammation, prevents neuronal metabolic dysfunction and synaptic degeneration, and enhances cognitive function. These effects were inhibited by [D-Lys (3)]-GHRP-6 (*Zhao et al., 2017*). |
| Aβ deposition | | Ghrelin attenuates oxidative damage and improves cognition by inhibiting IKK/NF-κB/BACE1-mediated Aβ production. These effects were inhibited by [D-Lys (3)]-GHRP-6 (*Ma et al., 2020*). |
| Neuroprotection | | Ghrelin improves the ultrastructural integrity of the hippocampal CA1 region, neuronal morphology, and cognition (*Shen et al., 2016*). |
| Oxidative stress | The DCI model was induced with streptozotocin in conjunction with a high-fat and high-sugar diet | Decreased levels of ghrelin and GHS-R1a in the gastric, serum, and hippocampal tissues of DCI rats result in oxidative damage and downregulation of the AMPK-PGC-1α-UCP2 signaling pathway, ultimately contributing to cognitive impairment (*Zhang et al., 2023a*). |
| Anti-inflammatory and anti-apoptotic | High glucose-induced cells after addition of ghrelin | Inhibition of the TLR4/NF-κB pathway in a high glucose environment leads to inhibition of high glucose-induced apoptosis in PC12 cells (*Liu et al., 2013*). |
| Insulin sensitivity | | Modulating the PI3-K/Akt-GSK-3β pathway enhances glucose uptake, improves insulin sensitivity, alleviates insulin resistance, and diminishes aberrant tau phosphorylation (*Chen et al., 2010*). |

thereby playing a crucial regulatory role (Fig. 2). GHS-R1a is extensively transcribed in several crucial regions of the brain and is expressed by various immune cells and peripheral tissues (*Buntwal et al., 2019*). Ghrelin does not regulate glucose metabolism in GHS-R1a knockout (GHS-R KO) mice or mice treated with GHS-R1a antagonists. GHS-R1a protein levels were decreased in the DCI rat model. GHS-R1b can regulate GHS-R1a in a heterodimeric form (*Taub, 2008*), but how it affects cognitive function remains unclear.

## Association of ghrelin with DCI-related pathologies

The pathogenesis of DCI is intricate, and it is currently believed that the pathogenesis of DCI mainly includes insulin resistance due to disorders of glucose and lipid metabolism, which in turn affects multiple signaling pathways to promote the deposition of β-amyloid (Aβ) and the formation of neural protofiber tangles (NFTs) (*Przybycien-Gaweda et al., 2020*). Simultaneously, the sustained elevation of blood glucose induces impaired glucose metabolism and disrupts cerebral energy homeostasis, thereby resulting in increased oxidative stress and mitochondrial dysfunction, altered synaptic plasticity, and augmented neuroinflammation that causes apoptosis and leads to diabetic cognitive impairment (*Dutta et al., 2022*). Each factor above is essential to the onset and progression of DCIs.

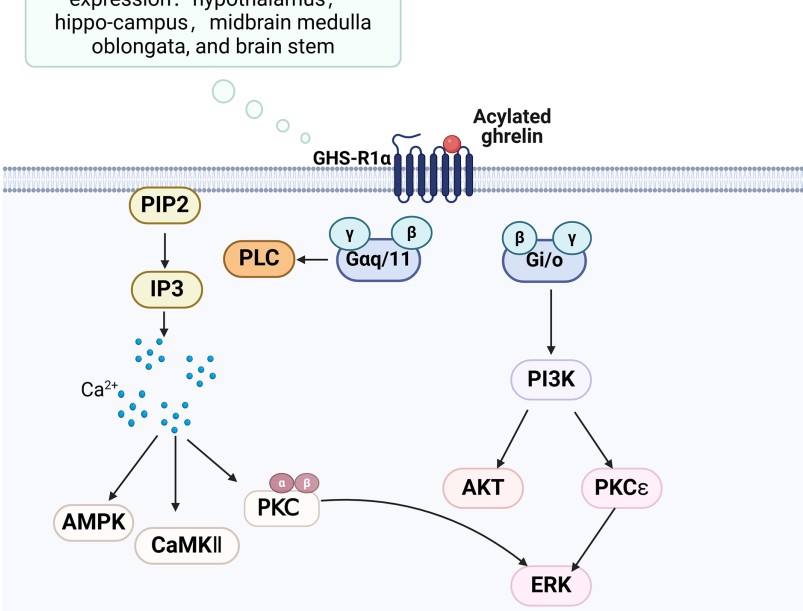

**Figure 2 Ghrelin binds to GHS-R and activates related pathways.** Created in Biorender.

## Ghrelin and Aβ

Aβ peptides are generated through the amyloidogenic pathway after amyloid precursor protein (APP) is sequentially proteolytically cleaved by β- and γ-secretase enzymes. Meanwhile, aggregation and aberrant accumulation of Aβ peptides lead to the formation of amyloid plaques, which can impact cognitive functions (*Allinquant, Clamagirand & Potier, 2014*). T2DM patients were found to have Aβ deposits in the brain and exhibit impairment of the insulin signaling pathway (*Anbari & Al-Harithy, 2023*). The impact of insulin imbalance on Aβ formation has been demonstrated. For instance, high insulin levels can compete with Aβ for binding to insulin-degrading enzyme (IDE), thereby influencing Aβ clearance. Meanwhile, Aβ oligomers can trigger Aβ amyloid deposition by promoting IRS-1 phosphorylation and undermining insulin signaling (*Clarke et al., 2015*; *Messier & Teutenberg, 2005*). They can also accelerate tau hyperphosphorylation through activation of GSK3β, thereby contributing to the development of DCIs.

It has been demonstrated that the use of ghrelin inhibits the increase of Aβ-positive cells in the hippocampus of diabetic rats and hippocampal progenitor cells in a high glucose culture. Ghrelin also mitigates Aβ accumulation by upregulating the levels of Aβ precursor-activated protein phosphatase 1 (PP1) in the hippocampus of STZ-induced diabetic rats, inhibiting the activation of NF-κB and IKKβ, and activating the AMPK pathway within the brain, thereby delaying cognitive decline. Injections of acyl ghrelin into Aβ$_{1-40}$ model mice and ghrelin knockout mice demonstrated that both groups exhibited enhanced recognition of novel objects; however, this treatment did not influence memory function in normal mice (*Santos et al., 2017*). It has been suggested that acyl ghrelin

exhibits memory-protective effects exclusively in the injury state; however, further research is necessary to determine the efficacy of long-term ghrelin use. Another study complicates the relationship as [D-Lys (3)] GHRP-6, a GHSR antagonist, demonstrated reduced hippocampal Aβ levels in obese rats while simultaneously lowering plasma cholesterol and glucose (*Madhavadas, Kutty & Subramanian, 2014*). It has been suggested that diabetic memory impairment induced by Aβ deposition is a key aspect of ghrelin intervention (Fig. 3). However, the precise role of ghrelin in the series of subsequent damages caused by Aβ remains to be elucidated.

## Ghrelin and mitochondrial dysfunction

Mitochondrial dysfunction is a critical factor contributing to the development of DCI (*Luo et al., 2022*). Prolonged elevation of blood glucose levels can lead to an increase in ROS, which disrupts the oxidative phosphorylation process within the mitochondrial electron transport chain, thereby diminishing ATP production. This reduction in ATP further exacerbates ROS accumulation and results in impaired energy supply to the hippocampus (*Chung, Kim & Song, 2022*). The imbalance between mitochondrial division and fusion in the hippocampus is exacerbated, leading to an increased phosphorylation level of Drp1 (*Hu et al., 2022*; *Maneechote et al., 2022*), which promotes mitochondrial fission. GSK3β can activate this process and may result in synaptic dysfunction. It also disrupts mitochondrial biosynthesis in the hippocampus by reducing the expression levels of PGC-1a mRNA and protein and decreasing the copy number of mitochondrial DNA (mtDNA). This impairment ultimately leads to a decline in cognitive functioning (*Panes et al., 2022*; *Zhao et al., 2021*).

Ghrelin effectively improves mitochondrial dysfunction through various mechanisms. In the DCI rat model, ghrelin levels were decreased and accompanied by a reduction in the expression of AMPK-PGC-1a-UCP2, a mitochondria-associated protein, in the hippocampus. The ghrelin analog (Hex) mitigates β-cell damage in diabetic rats by enhancing the Bax to Bcl-2 ratio, inhibiting the activation of caspase-3 and caspase-9, and safeguarding mitochondrial function and structure. Ghrelin also protects rat metazoan cells and hypothalamic neurons against mitochondrial membrane depolarization induced by Aβ oligomers (*Gomes et al., 2014*; *Martins et al., 2013*). Ghrelin promotes mitochondrial autophagy and facilitates the translocation of PINK1/Parkin to damaged mitochondria, thereby restoring them through activation of the AMPK/SIRT1/PGC-1a pathway in a model of rotenone-induced SH-SY5Y cell injury (Fig. 4) (*Eid, 2024*). The findings suggest that ghrelin may mitigate damage to diabetic β-cells and exert a neuroprotective effect by enhancing mitochondrial function.

## Ghrelin and synaptic dysfunction

DCI is closely linked to altered synaptic plasticity. The impaired ability to induce synaptic enhancement in the hippocampus of diabetic rats, resulting from chronic hyperglycemia, is associated with the reorganization of postsynaptic receptors involved in glutamatergic synaptic transmission (*Zhong et al., 2019*). This process entails reduced expression of various signaling molecules, such as neurotrophic factors and synapse-associated proteins

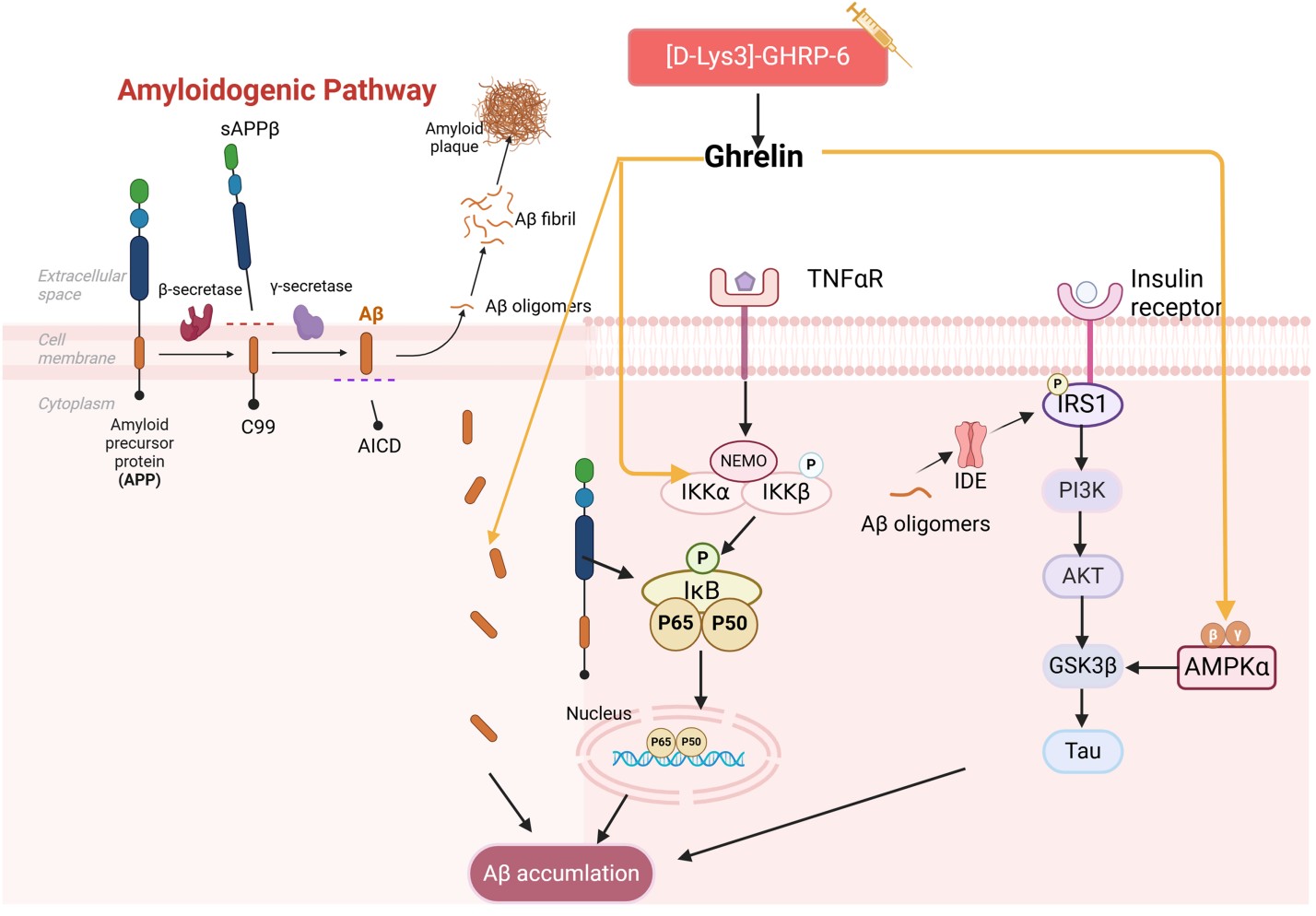

**Figure 3 Effect of ghrelin on Aβ homeostasis.** Created in Biorender.

and diminished release of transmitters (AMPAR and NMDAR) from glutamatergic neurons (*Ortiz et al., 2022*). Consequently, this inhibition of long-term potentiation leads to cognitive dysfunction.

On the one hand, ghrelin regulates the release of neurotrophic factors such as pro-NGF, mature NGF, and BDNF in diabetic rats, thereby enhancing synaptophysin expression and promoting dendritic spine formation. Additionally, it enhances the long-term potentiation of Schaffer's lateral branch-CA1 synapses, ultimately improving cognitive function (*Hornsby et al., 2020*). On the other hand, even in the presence of an NR2B-IR antagonist, ghrelin significantly elevated the expression of NR2B-IR and effectively reversed the memory suppression induced by this antagonist (*Ghersi et al., 2015*). The GHS-R1a protein is expressed near excitatory synapses in the hippocampus. It enhances synaptic transmission by regulating the transport of AMPARs and facilitating two distinct forms of long-term potentiation in this brain region. The above experiments demonstrate that ghrelin and GHS-R can activate multiple processes of synaptic transmission in the hippocampus and impact learning and memory function. However, synaptic transmission

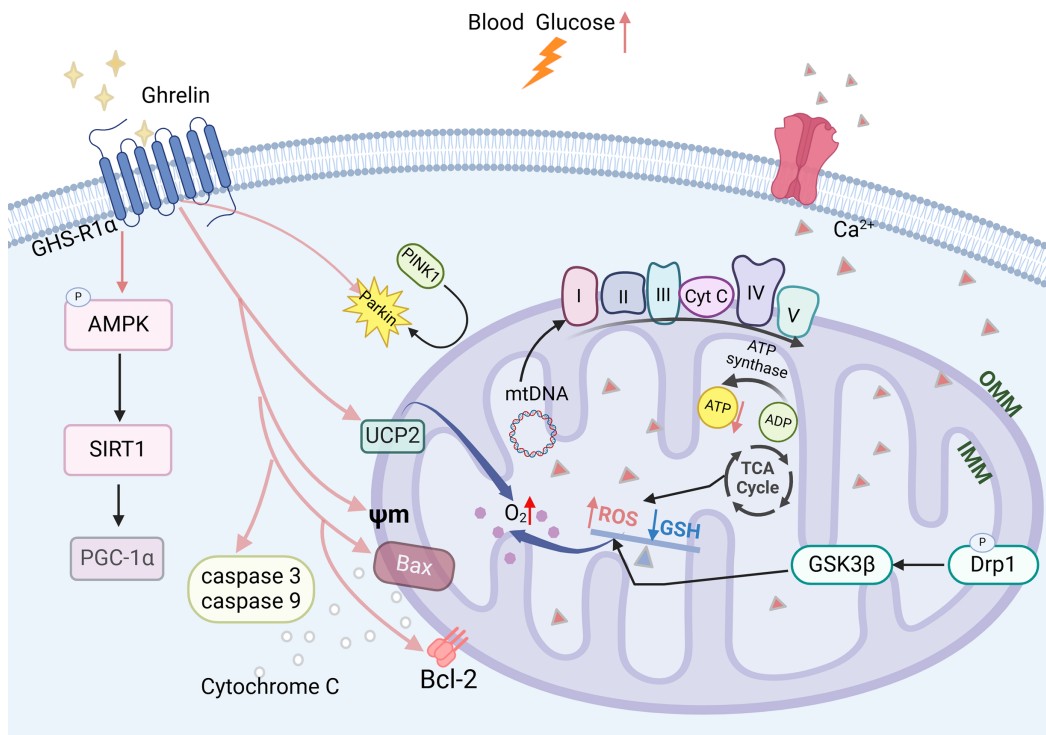

**Figure 4 The effect of ghrelin on mitochondrial dysfunction.** Created in Biorender.

and synaptic plasticity are intricate processes involving the transmission of information through multiple protein molecules, and it is not feasible to indefinitely augment this process in response to ghrelin stimulation. It has been demonstrated that ghrelin is limited in enhancing memory mediated by synaptic transmission (*Stoyanova & le Feber, 2014*). All the examples mentioned above demonstrate the importance of hormonal regulation in maintaining brain homeostasis (Fig. 5).

## Ghrelin and neuroprotection
Neuroinflammation has been identified as a potent driver of DCI progression. In a diabetic rat model, persistent diabetes leads to the onset of chronic inflammation and elevated levels of inflammatory factors. These factors activate neuroglia and induce the secretion of various pro-inflammatory factors, resulting in damage to hippocampal neurons and subsequent cognitive dysfunction (*Yang & Zhou, 2019*). The activation of astrocytes, microglia, and various other types of glial cells primarily drives neuroinflammation. Consequently, maintaining the stability of glial cells is crucial to ensuring neuronal homeostasis.

### *Microglia*
The initial indication of neuroinflammation is the activation of microglia. Prolonged elevated sugar levels result in excessive microglia activation, heightened central release of pro-inflammatory cytokines, and increased Aβ deposition, leading to synaptic dysfunction,

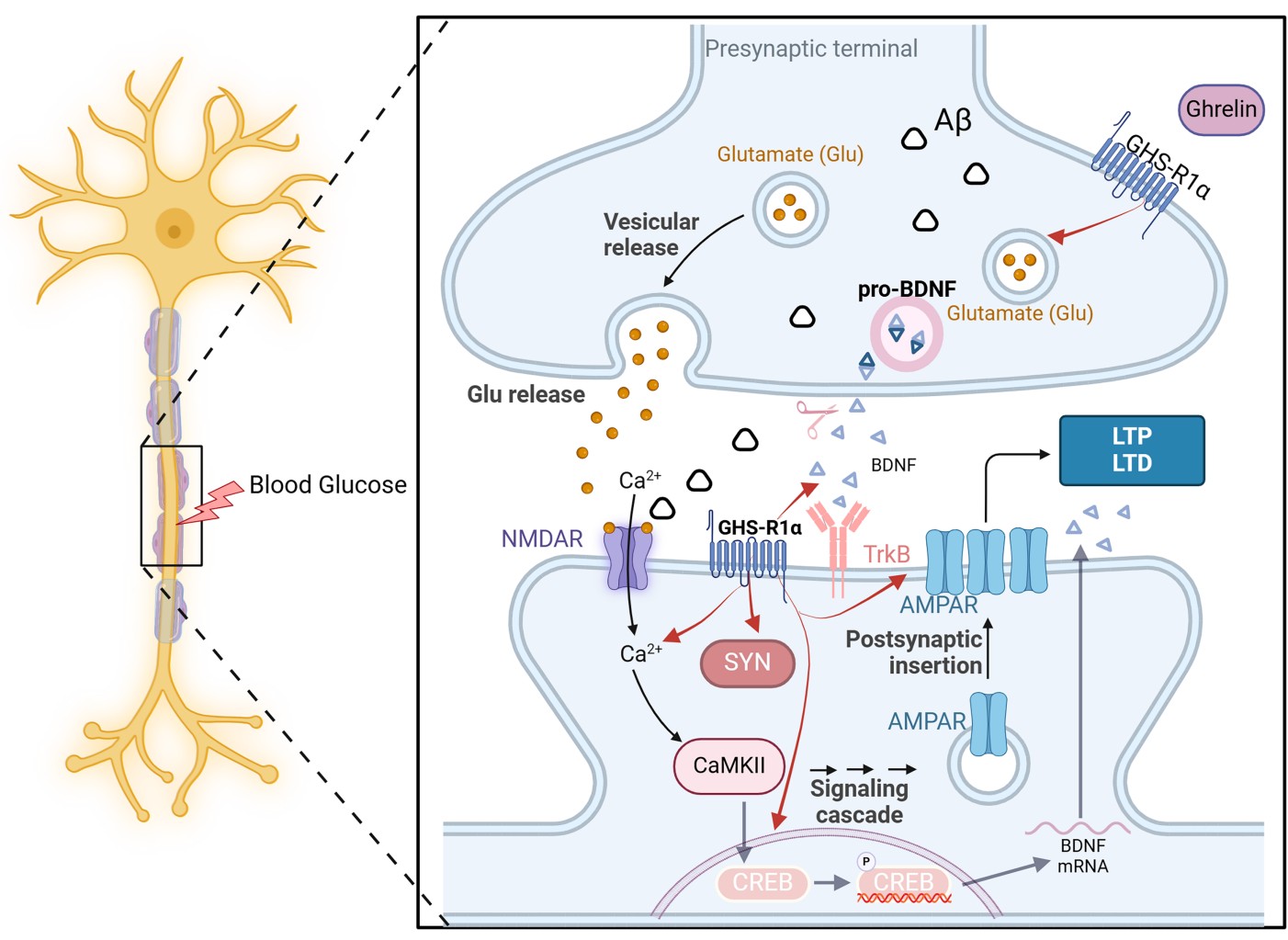

**Figure 5 The effect of ghrelin on synaptic dysfunction.** Created in Biorender.

ultimately culminating in neuronal degeneration and impairment of cognitive function (*Yin et al., 2017*).

Ghrelin enhances the behavioral deficits induced by diabetes by improving the impaired autophagic flux in BV2 cells while simultaneously inhibiting the activation of NLRP3 inflammatory vesicles and NF-κB signaling pathways (*Han et al., 2022*). The administration of ghrelin mitigated the activation of c-JNK in microglia stimulated by LPS and downregulated the expression of MAC-1, a marker for microglia, in neurons (*Koshenov et al., 2020*). The Cluster of Differentiation 36 (CD36) has been reported as an inflammatory mediator of Aβ, while growth hormone-releasing peptide agonists inhibit microglial cell inflammation and reduce neuronal death by binding to CD36. However, the proinflammatory effect of the activated state of microglia is not absolute. Different cellular phenotypes produce different inflammatory effects, which are determined by the type of microRNAs in microglia. Targeting different microRNA types can provide insight into the mysteries of microglial status and inflammatory responses.
### Oligodendrocytes

In mice with diabetes and concurrent cognitive impairment, a hyperglycemic environment adversely impacts the quantity and functionality of oligodendrocyte precursor cells (OPCs) and oligodendrocytes (OLGs), thereby hindering the process of white matter repair and resulting in myelin damage (*Wang et al., 2022*; *Yatomi et al., 2015*). OLGs are specialized myelinating cells responsible for forming axonal myelin. They facilitate rapid and precise conduction of electrical signals by maintaining axonal action potentials. Additionally, oligodendrocytes secrete various factors that provide trophic support to neurons, regulate synaptic development, and play a crucial role in central nervous system (CNS) injury repair (*Nasrabady et al., 2018*).

Mature myelinating oligodendrocytes expressing GHS-Rs undergo degeneration and apoptosis in response to $H_2O_2$ stimulation. The binding of ghrelin to these cells partially inhibits the p38/MAPK signaling pathway and increases the level of p-ERK, thereby protecting against apoptosis (*Yune et al., 2007*). p38/MAPK signaling during microglial activation mediates the production of the neurotrophic factor precursor pro-NGF to promote oligodendrocyte apoptosis (*Lee & Yune, 2014*). When LPS-induced BV2 cells are cocultured with oligodendrocytes, ghrelin suppresses the activation of microglial cytokines induced by LPS and mitigates damage to oligodendrocytes. The findings mentioned above indicate that ghrelin mitigates inflammation and oxidative damage in oligodendrocytes, thereby contributing to the amelioration of myelin pathology associated with neurodegenerative diseases.

### Astrocytes

Astrocytes are the most abundant type of glial cells in the brain, playing crucial roles in supporting and facilitating neuronal growth. They are involved in glycogen storage, contribute to glucose metabolism, and help maintain the integrity of the blood-brain barrier. Astrocytes are believed to play a pivotal role in facilitating neuron-astrocyte communication, actively driving the bi-directional interactions between pre- and postsynaptic neurons (*Lin, Zheng & Zhang, 2018*). Conversely, diabetes triggers the activation of brain-reactive astrocytes, reduces glycogen levels in astrocytes, increases lactate concentrations, disrupts glucose-energy metabolism, diminishes glutamate uptake, and dismantles the coupling between astrocytes and neurons. These alterations contribute to the development of DCI (*Meng et al., 2023*; *Shen et al., 2023*).

In a model of traumatic brain injury, ghrelin prevents the production of proinflammatory cytokines and ROS by astrocytes and microglia, thereby attenuating neurodegeneration and blood-brain barrier disruption caused by apoptosis (*Ergul Erkec et al., 2024*; *Frago & Chowen, 2017*). In the STZ-induced diabetes model, ghrelin reduces astrocytic GFAP immunoreactivity in the rat brain and maintains the balance between pro-NGF and NGF, thereby protecting hippocampal neurons. Additionally, it stimulates an increase in astrocyte populations by binding to the GHS-R1a receptor and activating the PI3K/AKT signaling pathway, thus playing a neuroprotective role (*Baquedano et al., 2013*). In addition, ghrelin plays a crucial role in modulating glucose uptake by astrocytes and influences the expression of proteins involved in lactate and glutamine metabolism within

these cells (*Chowen, Frago & Fernández-Alfonso, 2019*). Conversely, the administration of ghrelin receptor antagonists inhibited these beneficial effects, thereby underscoring the significance of ghrelin in mitigating astrocyte dysfunction.

### NG2-glial

NG2-glial, formerly referred to as oligodendrocyte precursor cells (OPC), can differentiate into oligodendrocytes. Under specific conditions, they can also give rise to astrocytes and neurons. These processes are crucial for sustaining oligodendrocyte production and ensuring myelin plasticity. NG2-glial, the only class of glial cells known to establish direct synaptic connections with neurons and receive synaptic inputs from them (*Du et al., 2021*), play a vital role in preserving both the structural and functional integrity of the blood-brain barrier (BBB).

The presence of myelin deficiency is significantly correlated with cognitive impairment in individuals with DM. Some studies have identified NG2-glial in the brain as a potential source for myelin repair. Furthermore, it has been demonstrated that the expression of GHS-Rla was detected in mouse brain neural stem cells (mbNSCs) after seven days of *in vitro* culture. Notably, stimulation with ghrelin resulted in an increase in the number of NG2-glial within mbNSCs (*Gong et al., 2020*). However, it has been demonstrated that hippocampal NG2-glial in db/db mice contribute to the disruption of the blood-brain barrier in diabetes through the secretion of MMP-9 (*Li et al., 2024*). The discussion of the involvement of ghrelin and NG2 cells in diabetes and the central nervous system is warranted.

Neurons do not function autonomously; instead, they rely on glial cells to facilitate signal exchange in response to feedback from changes in the brain environment (Fig. 6). However, this feedback process is also modulated by other proteins and factors, necessitating further experimental evidence to substantiate the impact of ghrelin on DCI-induced glial cells.

## ANALOGS

Ghrelin, a type of peptide, inherently presents several disadvantages, including low stability, a short plasma half-life, and poor bioavailability. These limitations significantly restrict its clinical applications. Consequently, in recent years, numerous researchers have begun to explore ghrelin analogs and peptides with similar functions.

Compared with ghrelin, the gastric starvation hormone analog Dpr³ghrelin (Dpr³ghr), which is more stable and has prolonged action, has been shown in *in vitro* experiments to increase SHSY-5Y cell viability and mitigate methylglyoxal-induced SHSY-5Y cytotoxicity through pretreatment with Dpr3ghr. Additionally, it may reduce the Bax/Bcl-2 ratio by modulating the PI3K/AKT pathway, thereby decreasing apoptosis and demonstrating potential neuroprotective properties (*Popelová et al., 2018*).

MK0677 is a potent, high-affinity, orally active, nonpeptide agonist of the GHS-R. It has been demonstrated to improve Aβ deposition, neuroinflammation, and neurodegeneration when administered intraperitoneally (*Jeong et al., 2018*). Oral administration of IGF-1 increases serum levels in hemodialysis patients and those with AD. Moreover, MK0677

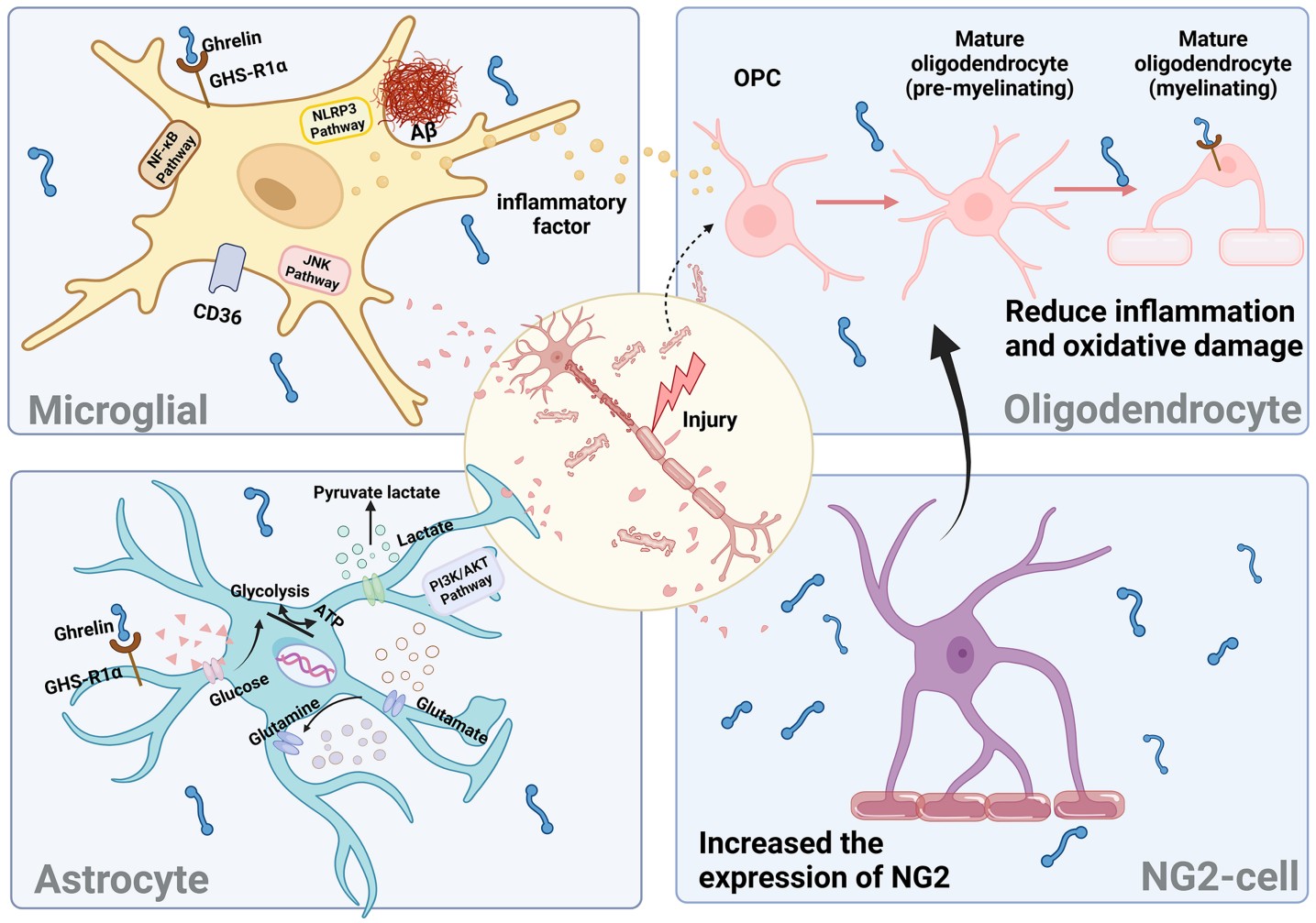

**Figure 6** **The effect of ghrelin on neuroinflammation.** Created in Biorender.

only enhanced hippocampal neurological function; however, no preventive or ameliorative effects on Aβ deposition, synaptic loss, microglial activation, or cognitive deficits were observed. Variations in the administered doses and methods of Aβ detection may be responsible for these differences, but further investigation is needed to determine whether these variations have a cognitive-improving effect (*Tian et al., 2019*).

LY444711 has been shown to improve cognitive deficits in APPSwDI transgenic mice, reduce Aβ deposition and microglial activation, and improve the pathological manifestations of AD (*Dhurandhar et al., 2013*). However, its effectiveness for other pathogenic mechanisms of cognitive deficits and its pathways of action need to be further explored.

Ghrelin is a class of appetite-stimulating hormones, whereas nesfatin-1 is an anorexigenic hormone produced in the hypothalamic nuclei. These two hormones play distinct roles in regulating the body's energy metabolism and cardiovascular system; however, their functions in neuroprotection appear to be similar. Nesfatin-1 also influences glucose metabolism through the GHS-R pathway and plays a significant role in

regulating mood and cognitive function. Intracerebral injection of nesfatin-1 in rats subjected to a high-fat diet significantly enhanced insulin signaling by modulating the AKT/AMPK/TORC2 signaling pathway (*Yang et al., 2012*). Nesfatin-1 plays a crucial role in the pathogenesis of dysregulated glucose and lipid metabolism, as well as neurobehavioral disorders observed in NAFLD rat models. Clinical studies indicate that plasma nesfatin-1 may be linked to the onset of cognitive dysfunction in individuals with T2DM comorbidities (*Xu et al., 2021*). However, due to the limited number of relevant experimental studies and small sample sizes, a substantial amount of future research is necessary to validate the correlation between these two factors.

## CONCLUSION

In recent years, a growing body of evidence has revealed the significant role of the gut-brain axis (GBA) in the pathophysiology of central nervous system complications (*Góralczyk-Bińkowska, Szmajda-Krygier & Kozłowska, 2022*). Ghrelin, a class of gut-brain peptides, plays a dual role in enhancing peripheral glucose metabolism and central cognitive function. Several studies have indicated that patients with diabetes who also experience cognitive impairment exhibit low levels of ghrelin. Furthermore, it has been suggested that regulating ghrelin levels may delay the pathological symptoms associated with DCI. This finding positions ghrelin as a promising target and potential predictor for the treatment of DCI. Therefore, this review utilizes ghrelin as a focal point to elucidate the efficacy and potential of ghrelin's intervention in various pathogenic mechanisms associated with DCI. This approach stimulates interest among researchers to expand therapeutic options and diagnostic criteria for DCI. Nonetheless, the existing research on the association between ghrelin and cognitive impairment in DM remains primarily confined to fundamental experimental studies, thereby presenting a broad scope for further exploration and promising developmental opportunities from this vantage point. Nevertheless, natural ghrelin has a short plasma half-life and poor stability, which hinders its development. While ghrelin mimetics and receptor agonists have addressed these limitations, most studies involving mimetics focus on conditions such as diabetic gastroparesis and cachexia, with comparatively fewer investigations into diabetic brain complications that warrant further exploration. In addition, while existing studies have primarily concentrated on acylated ghrelin and GHS-R1a, it is important to recognize that deacylated ghrelin and GHS-R1b also hold significant potential for further investigation. Deacylated ghrelin constitutes a relative content of approximately 60–90% of the total growth hormone-releasing peptide pool, making it the most stable and slowly metabolized form of ghrelin present in plasma (*Airapetov et al., 2021*). Deacylated ghrelin can independently participate in the regulation of various bodily systems, distinct from the GHS-R1a pathway, or modulate the metabolic activity of the growth hormone-releasing peptide system, unlike acylated ghrelin. It has been demonstrated to protect against muscle atrophy, safeguard cardiac function, and influence insulin levels (*Liang et al., 2021*). GHS-R1b can also penetrate the brain through passive diffusion; however, its mode of action and effects within the central nervous system remain unknown. Future research should focus on elucidating the mechanisms of deacylated ghrelin and GHS-R1b to expand therapeutic

options for DCI. Some studies have indicated that ghrelin may have potential adverse effects on certain metabolic diseases and could exacerbate the progression of these conditions, thereby complicating the issue further.

The limited sample size, interindividual variations, and inadequate clinical data have contributed to the lack of clarity regarding the impact of ghrelin and GHS-R on DCI pathology as well as the specific pathways of action, signaling cascades, and molecular mechanisms involved. Therefore, future investigations should prioritize correlative studies examining the effects of ghrelin on DCI pathology and elucidating the precise mechanisms underlying downstream pathways.

### Funding
This work was supported by grants from the Shaanxi Provincial Department of Education Service Local Special Research Program Project (22JC029). The funders had no role in study design, data collection and analysis, decision to publish, or preparation of the manuscript.

### Grant Disclosures
The following grant information was disclosed by the authors:
Shaanxi Provincial Department of Education Service Local Special Research Program Project: 22JC029.

### Competing Interests
The authors declare that they have no competing interests.

### Author Contributions
- Yuhan Zhang conceived and designed the experiments, analyzed the data, prepared figures and/or tables, and approved the final draft.
- Ruihua Zhang conceived and designed the experiments, authored or reviewed drafts of the article, and approved the final draft.
- Xin Wang performed the experiments, prepared figures and/or tables, and approved the final draft.
- Leilei Shi analyzed the data, authored or reviewed drafts of the article, and approved the final draft.
- Hongzhe Zhu analyzed the data, authored or reviewed drafts of the article, and approved the final draft.
- Jiping Liu analyzed the data, authored or reviewed drafts of the article, and approved the final draft.

### Data Availability
This is a literature review.

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
