# Peer review of "Potential role of ghrelin in neuroprotection and cognitive function: implications for diabetic cognitive impairment"

_PeerJ, doi:10.7717/peerj.18898_

## Round 0.1 · original submission · Major Revisions

The manuscript is overly long and challenging to read, with excessive citations. The authors should condense sections, revise the highlights for clarity and conciseness, and include a comprehensive, critically summarized bibliography with a summary table.

Please address comments from all of the reviewers in a point wise manner.

·

Basic reporting

In the references you should try to standardize the section of authors, which should appear entirely in English.

Experimental design

The methodology and presentation of the search results are adequate.

Validity of the findings

The novelty of the information presented is adequate, the graphics are clear and the importance of the results makes this manuscript publishable.

Additional comments

Dear editors, I thank you for the invitation to review this excellent manuscript.

·

Basic reporting

The review manuscript entitled “Ghrelin in diabetic cognitive dysfunction: Pathologic roles and therapeutic implications” has been evaluated.
They stated that "this article offers a comprehensive review and future outlook on the potential connections between the impact of ghrelin on diabetic cognitive impairment and the development prospects of ghrelin mimics"
They aimed to present novel targets and concepts for preventing and treating of Diabetic Cognitive impairment
They should include references that support their specific issue: e.g.
• Ghrelin levels were found to be significantly lower in patients with type 2 diabetes mellitus (T2DM) combined with mild cognitive impairment (MCI) than in normal subjects Logistic regression analysis revealed that fluctuations in ghrelin levels may be a contributing factor independently impacting MCI in patients with T2DM.
The title of the manuscript does not closely match with the content of the manuscript. Its focuses on ghrelin in mental cognitive or mental disorders rather than type 2 diabetes mellitus-induced cognitive improvements.

They stated that “(Line 193-195); “Furthermore, there was a decrease in the copy number of mitochondrial DNA (mtDNA), which may be one of the contributing factors to the occurrence of DCI [41-43] (Fig. 4)”
However, I could not analyze the references because all of them were Chinese. In addition, Figure 4 may not support their suggestion.

The major confusing problem for me is the references presented in Chinese. Thus, I can not check the correctness
They prepared a manuscript with nice, valuable information. The weakness was that they did not mention the sister hormone of ghrelin, called as nesfatin-1, which showed importance in diabetes and cognitive impairments.
After correction, it could be acceptable.
Best regards.

Experimental design

The major confusing problem for me is the references presented in Chinese. Thus, I can not check the correctness
They prepared a manuscript with nice, valuable information. The weakness was that they did not mention the sister hormone of ghrelin, called as nesfatin-1, which showed importance in diabetes and cognitive impairments.

Validity of the findings

The title of the manuscript does not closely match with the content of the manuscript. Its focuses on ghrelin in mental cognitive or mental disorders rather than type 2 diabetes mellitus-induced cognitive improvements.

Additional comments

after correction it could be acceptable

Reviewer 3 ·

Basic reporting

In this manuscript, zhang et al., provide an overview of Ghrelin in diabetic cognitive dysfunction: Pathologic roles and therapeutic implications,It is to be recognized that the authors successfully managed to outline many of the implicated molecular mechanisms and provided well-designed and very informative figures. Nevertheless, there are certain issues that lower the overall quality of the manuscript and aspects that the authors could further work on to improve the readability, timeliness, and relevance of it.

Experimental design

- Overall, the manuscript is very long and at times difficult to read. The authors could shorten some sections and instead of citing almost the entirety of the relevant literature, could provide a more concise and critical summary that would help the less expert readers to grasp the main points.
- Highlights should be significantly revised and adapted to be indeed highlights with short, direct, specific and clear information.
-a complete revision of available bibliography should be included in the manuscript and summary table.

Validity of the findings

The effect of Ghrelin on specific pathologies is somehow limited and should include information from recent studies and those covering the effects on patients. Relevant studies that could significantly strengthen this review seem to be missing regarding the effects of Ghrelin in diabetic cognitive dysfunction pathology and related alterations.

Additional comments

- The structure of the manuscript seems appropriate, although some concepts and parts should be revised to avoid duplications and similar/overlapping messages.
-The citation of Chinese literature was avoided as much as possible.
-There is too much text in the Table and it should be condensed.
-The quality of the figures is poor and should be improved.

---

## Round 0.2 · accepted · Accept

In my opinion, the authors have addressed all of the reviewers' comments and this manuscript is ready for the publication.

Reviewer 3 ·

Basic reporting

no comment

Experimental design

no comment

Validity of the findings

no comment

Additional comments

My concerns have been addressed in the revised version.